# The influence of topical non-steroidal anti-inflammatory drugs on the intraocular pressure lowering effect of topical prostaglandin analogues—A systemic review and meta-analysis

**Kang-Jung Lo[1], Yu-Chieh Ko[1,2], De-Kuang Hwang**[1,2]*, **Catherine Jui-Ling Liu**[1,2]

**1** Department of Ophthalmology, Taipei Veterans General Hospital, Taipei, Taiwan, **2** Faculty of Medicine, National Yang-Ming University School of Medicine, Taipei, Taiwan

* m95gbk@gmail.com

## Abstract

**Data Availability Statement:** All relevant data are within the manuscript and its Supporting Information files.

### Purpose

This study aimed to review previous articles and evaluate the influence of topical non-steroidal anti-inflammatory drugs (NSAIDs) on intraocular pressure (IOP) in glaucoma patients who were treated with prostaglandin analogues (PGs).

### Method

The presenting study was designed as a meta-analysis of previous research. Databases include PubMed, Web of science, Cochrane library, and Embase were searched with keywords of "intraocular pressure, prostaglandin analogues, NSAIDs, latanoprost, travoprost, bimatoprost, tafluprost, unoprostone, latanoprostene bunod, ketorolac, diclofenac, nepafenac, bromfenac, flurbiprofen". Inclusion criteria were: 1. Study population were glaucoma patients; 2. Comparison between PGs monotherapy and PGs in combination with topical NSAIDs; 3. Changes of IOP as final outcomes. Studies with non-randomized design, treatments combining other anti-glaucomatous drugs, or unavailable absolute IOP were excluded from the analysis. Estimated difference in IOP were calculated using STATA 14.0.

### Result

Seven studies were retrieved for this meta-analysis. Since there is a significant heterogeneity ($I^2 = 94\%$) in these studies, random-effect model was used to calculate pooled standardized mean differences (SMD). Our results showed a significantly favorable IOP lowering effect in glaucoma patients treated with combination of topical NSAIDs and PGEs (SMD: 1.3 and -0.03, 95% CI: 0.29 to 2.38 and -0.32 to 0.26, Z = 2.50 and 0.23, p = 0.013 and 0.820, respectively).

**Funding:** The author(s) received no specific funding for this work.

**Competing interests:** The authors have declared that no competing interests exist.

## Conclusion

Results of our meta-analysis suggested that topical NSAIDs may enhance the IOP lowering effect of topical PGs in glaucoma patients.

## Introduction

Glaucoma is a progressive optic neuropathy that can cause irreversible loss of vision [1]. Several risk factors are associated with the development of glaucoma, of which only high intraocular pressure (IOP) and large diurnal fluctuations in intraocular pressure can be manipulated [1, 2]. Therefore, the mainstay of treatment for this disease focuses on controlling these factors. Prostaglandin (PG) analogues are well-known topical anti-glaucoma medications, with excellent potency in reducing IOP, good circadian IOP control and few side effects [3–5]. PG analogues reduce IOP by increasing uveoscleral outflow, which involves relaxation of ciliary muscles and remodeling of the extracellular matrix within the ciliary muscles and sclera [6].

Topical nonsteroidal anti-inflammatory drugs (NSAIDs) are used clinically to reduce postoperative ocular inflammation, prevent macula edema after cataract surgery, and maintain intraoperative mydriasis during cataract surgery [7–10]. As a cyclooxygenase (COX) inhibitor, NSAIDs inhibit the production of PGs [11]. Since the underlying mechanism of these two drugs are theoretically opposite, there were some debates regarding whether we should avoid NSAID or discontinue PG if a glaucoma patient suffers from macular edema after surgery. Patient's intraocular pressure could become relatively hard to be controlled if we discontinue PG. On the other hand, long-term macular edema would result in visual decline in patients. It is important to clarify whether IOP control is affected by concomitant treatment with topical NSAIDs and PG analogues in glaucoma patients or not [12–17]. Therefore, the aim of this study was to review previous articles and evaluate the influence of topical NSAIDs on IOP in glaucoma patients being treated with PG analogues via a literature review and meta-analysis.

## Methods

### Literature search

A literature search of PubMed, ISI Web of Science, EMBASE, and Cochrane library databases was performed to identify relevant studies. The search combined keywords related to NSAIDs (ketorolac, diclofenac, nepafenac, bromfenac, flurbiprofen), PG analogues (latanoprost, travoprost, bimatoprost, tafluprost, unoprostone, latanoprostene bunod) and IOP. Google Scholar and the websites of professional associations were also searched for information. Once relevant articles had been identified, their reference lists were also searched for additional articles. The final search was carried out in June 2020 without restricting the publication year, language, or methodology.

### Inclusion and exclusion criteria

We included publications that met the following inclusion criteria: (i) study design—randomized clinical trials (RCTs); (ii) population—patients with glaucoma; (iii) intervention—topical NSAIDs with PG analogues vs placebo vs PG analogues; and (iv) outcome variables—evaluating changes in IOP. Studies involving oral NSAIDs combined with PG analogues were excluded.

## Outcome measurements

Data on differences in the IOP between eyes treated with PG analogue monotherapy and PG analogues in combination with topical NSAIDs were obtained and analyzed. Positive numerals indicated that the IOP lowering effect of PG analogues was enhanced (i.e. the IOP in eyes treated with combination therapy was lower than the IOP in eyes treated with PG analogue monotherapy), and negative numerals indicated that the IOP lowering effect was reduced after adding topical NSAIDs.

## Study selection

After the literature searches had been combined and duplicates removed, the title and abstract of each unique article was systematically screened for eligibility. After applying the aforementioned inclusion/exclusion criteria, the full text of each article was read and analyzed. A flowchart illustrating the study selection process is shown in Fig 1.

## Data extraction

Two reviewers (KJL and DKH) extracted data from the eligible studies using a standardized paper form. Any discrepancies between the reviewers' results were resolved by consensus. For each eligible article, the first author, publication year, study location, study period, study design, baseline characteristics, treatment regimen, inclusion and exclusion criteria, and outcome were extracted.

## Quality assessment

Quality assessment was performed according to the risk-of-bias tool outlined in the Cochrane Handbook for Systematic Reviews of Interventions (version 5.1.0) [18]. Six key aspects that influence the quality of an RCT were assessed: sequence generation, allocation concealment, patient blinding, personnel and outcome assessors, management of incomplete outcome data, and completeness of outcome reporting, as well as other potential threats to validity. For each parameter, "yes" indicated a low risk of bias, "no" indicated a high risk of bias, and "unclear" indicated an unclear or unknown risk of bias.

## Statistical analysis

Stata 14.0 was used for the meta-analysis. The standardized mean difference (SMD) and 95% confidence interval (CI) were calculated. Heterogeneity between the results of different studies was examined using the $I^2$ statistic, and $P < 0.05$ and $I^2 > 50\%$ were considered to indicate statistically significant heterogeneity. If the included studies were not heterogeneous, the fixed-effects model was used for the analysis, otherwise a random-effects model was used.

# Results

## Identification of eligible studies

A total of 386 potentially relevant articles were identified, of which 375 were excluded because of duplication, animal studies, or in-vitro experimental studies, and the remaining 11 studies were retrieved for full text review. Two of these 11 studies which investigated oral NSAIDs were excluded [13, 19]. Among the remaining nine studies, seven focused on glaucomatous patients and two focused on healthy subjects. To unify the results, we also excluded these two studies on healthy subjects [20, 21]. The remaining seven full text articles were included in the final analysis [12, 14–17, 22, 23].

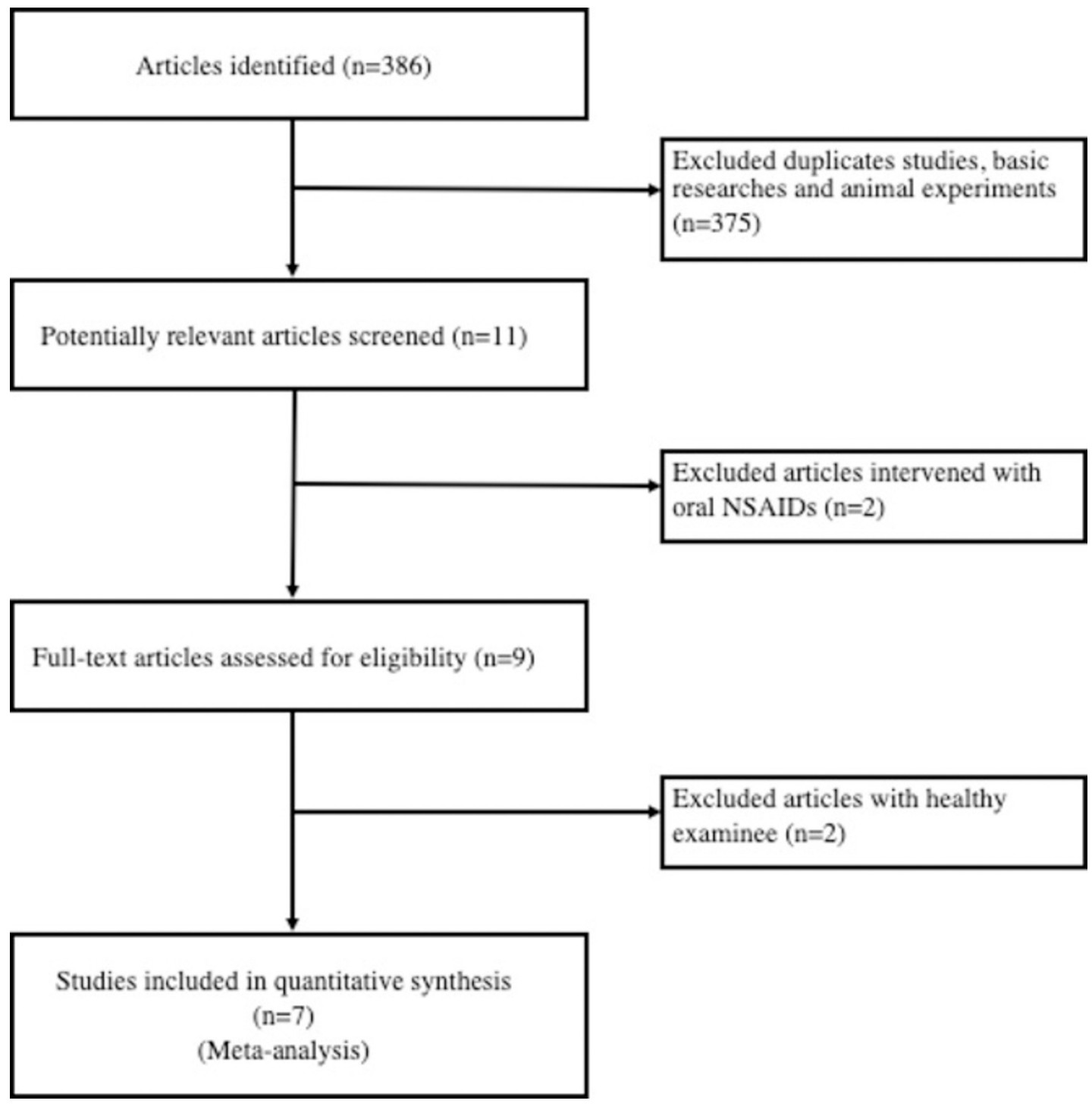

**Fig 1. Flowchart of publication search and selection.**

## Study characteristics

The main characteristics of the seven included RCTs are shown in Table 1. These studies were published between 2005 and June 2019 and were conducted in various countries: two in Italy,

**Table 1. Characteristics of the included RCT studies.**

| Author (year) | Journal | Location | Medication | | Type of patient | Primary endpoint | No. of eyes | | Sex (male/female) | | Age |
|---|---|---|---|---|---|---|---|---|---|---|---|
| | | | Glaucoma | NSAID | | | NSAID | Control | NSAID | Control | |
| C. Costagliola (2005) | Exp Eye Res | Italy | Timolol/ Latanoprost | Diclofenac | POAG[a] | 10 weeks | 32 | 32 | 16/16 | 16/16 | 55.4 ±5.11 |
| T. Chiba (2006) | Br J Ophthalmol | Japan | Latanoprost | Bromfenac | POAG/ OH[b] | 18 weeks | 13 | 13 | 6/7 | 6/7 | 65.2 ±8.8 |
| C. Costagliola (2008) | Curr Eye Res | Italy | Latanoprost | Ketorolac | POAG | 1 day | 16 | 16 | 9/7 | 9/7 | 59.45 ±4.55 |
| R. Sorkhabi (2011) | J Glaucoma | Iran | Latanoprost | Diclofenac | POAG | 2 weeks | 22 | 22 | 12/10 | 12/10 | 60.55 ±9.46 |
| E. Turan-Vural (2011) | Ophthalmologica | Turkey | Latanoprost/ Travoprost/ Brimatoprost | Ketorolac | POAG/ Pseudo-exfoliation glaucoma | 2 weeks | 30 | 30 | 15/15 | 15/15 | 60.8 ±11.5 |
| P. Özyol (2016) | J Glaucoma | Turkey | Latanoprost/ Travoprost/ Brimatoprost | Nepafenac | POAG | 2 weeks | 35 | 35 | 20/15 | 20/15 | 60.28 ±7.51 |
| Z. Simin (2018) | J Ophthalmol | China | Latanoprost | Pranoprofen | POAG | 10weeks | 24 | 24 | 13/11 | 10/14 | 32.4 ±4.7 |

[a]POAG = primary open angle glaucoma

[b]OH = ocular hypertension

two in Turkey, one in Japan, one in Iran, and one in China. The number of enrolled cases ranged from 13 to 35. There was no sex bias in these studies. The mean age of the participants in these studies ranged from 32.4±4.7 to 65.2±8.8 years. All of the participants received monotherapy with PG analogue eye drops before the experiment to reduce IOP fluctuations, except for the study by Costagliola in 2005 [22] in which the patients received topical 0.5% timolol for the first 2 weeks and then switched to PG analogues for the next 8 weeks. These patients were then further randomized into treatment and control groups. In the treatment group, all patients received NSAID eye drops in addition to PG analogues, while the control group only received PG analogue monotherapy.

## Quality and bias assessment of studies

The included RCTs had certain risks of bias, mainly the lack of blinding (Table 2). Sequence generation was appropriate in five studies, and allocation concealment was agreed in all

**Table 2. Results of quality and bias assessment of the included studies.**

| Author (year) | Sequence generation | Allocation concealment | Blinding | | Adequate assessment of each outcome | Selective reporting avoided | No other bias |
|---|---|---|---|---|---|---|---|
| | | | Patient | Assessor | | | |
| C. Costagliola (2005) | Yes | Yes | Yes | Yes | Yes | Yes | Yes |
| T. Chiba (2006) | Unclear | Yes | Yes | Yes | Yes | Yes | Yes |
| C. Costagliola (2008) | Yes | Yes | Yes | Yes | Yes | Yes | Yes |
| R. Sorkhabi (2011) | Yes | Yes | Yes | Unclear | Yes | Yes | Yes |
| E. Turan-Vural (2011) | Yes | Yes | Unclear | No | Yes | Yes | Yes |
| P. Özyol (2016) | Yes | Yes | Unclear | No | Yes | Yes | Yes |
| Z. Simin (2018) | Yes | Yes | Unclear | Unclear | Yes | Yes | Yes |

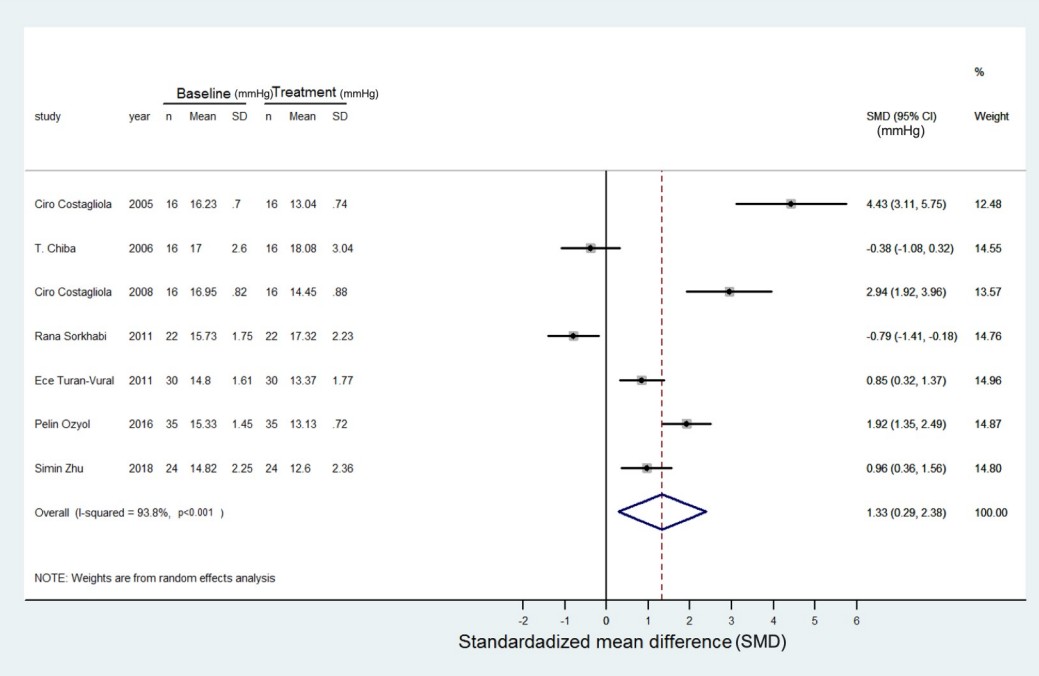

**Fig 2. Forest plot depicting the meta-analysis for the SMD (standardized mean difference) in IOP in the glaucoma patients treated with a combination of topical NSAIDs and PGs.** The SMD was calculated by Cohen method and the overall SMD favors the additional IOP lowering by this synergic effect. Most studies were treated with topical PGs alone for 4 weeks before combination therapy as baseline IOP, and the treatment IOP was measured as the peak reduction of the IOP in each study after the combination therapy. The treatment/baseline IOP was calculated in mmHg. The unit of SMD is in mmHg.

studies. Four studies clearly elaborated upon patient blinding, while only three studies clearly showed assessor blinding. All studies were judged to have a low risk of bias from selective reporting, because it was clear that all the main pre-specified outcomes had been reported.

## IOP outcome

Changes in IOP following concomitant treatment with NSAIDs and PG analogues were detailed in all studies. Five studies demonstrated positive values of IOP changes, indicating enhanced IOP reduction with added NSAIDs [12, 15, 17, 22, 23], while the other two reported negative values [14, 16]. There was significant heterogeneity ($I^2$ = 94%) among the studies. The random-effects model revealed a favorable IOP lowering effect in glaucoma patients treated with a combination of topical NSAIDs and PG analogues compared to PG analogue mono-therapy (SMD: 1.3 and -0.03 mmHg, 95% CI: 0.29 to 2.38 and -0.32 to 0.26, Z = 2.50 and 0.23, p = 0.013 and 0.820, respectively). The detailed forest plots are shown in Figs 2 and 3.

## Discussion

In this meta-analysis, we found that there was a further IOP lowering effect when adding additional NSAIDs to PG analogues compared to PG analogue monotherapy in glaucoma patients. Although topical NSAIDs are used clinically to reduce ocular inflammation or macula edema, no clear mechanism on how it may influence IOP has ever been reported.

The influence of topical NSAIDs on eyes with glaucoma under PG analogue therapy is still controversial. A reduced IOP lowering effect was reported by Chiba [14] and Sorkhabi [16],

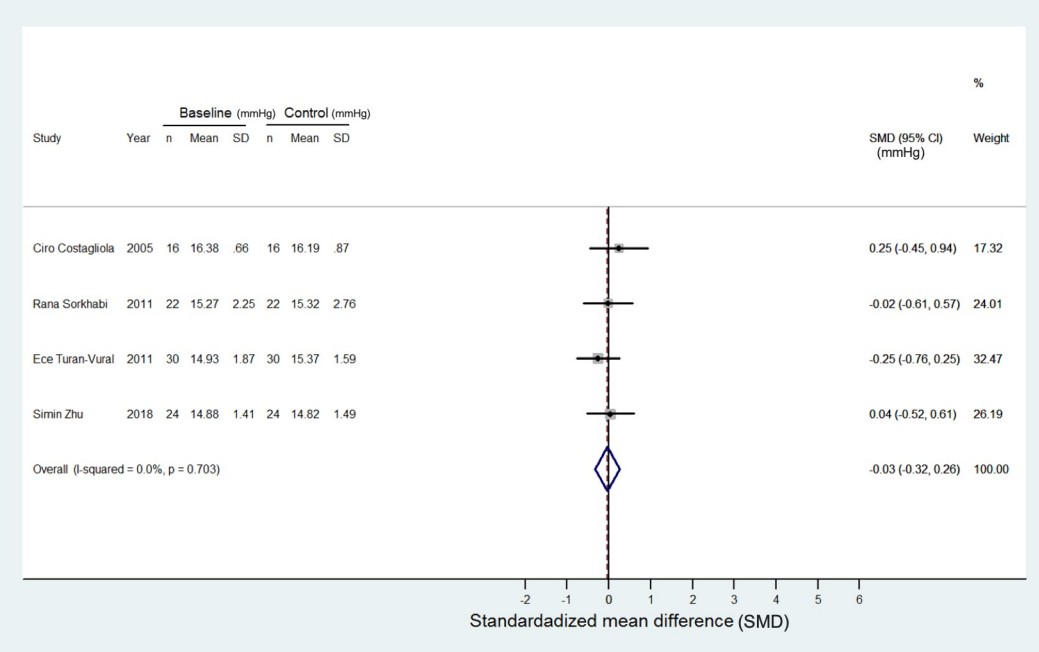

**Fig 3. Forest plot depicting the meta-analysis for the SMD (standardized mean difference) in IOP in the glaucoma patients treated with PG monotherapy.** The data only involved four articles because the other three articles did not provide precise data for calculations. The patients in this forest plot is the control group when comparing with patients in Fig 2. The SMD cross zero without treatment preference. The control/baseline IOP was calculated in mmHg. The unit of SMD is in mmHg.

who proposed that the induction of endogenous PGs by exogenous PG analogues was diminished after the application of additional NSAIDs. In contrast, an enhanced IOP lowering effect was reported when adding NSAIDs to PG analogues in the studies by Costagliola [12, 22], Ozyol [15], Turan-Vural [17], and Simin [23]. They proposed that NSAIDs can inhibit the production of endogenous PGs, thereby up-regulating receptor densities and causing a further reduction in IOP.

After adjusting the meta-analysis, our study still favored a synergistic effect when using NSAIDs and PG analogues simultaneously in glaucoma patients. We hypothesize that the mechanism of an enhanced IOP lowering effect when adding NSAIDs to PG analogues may be explained as follows. In the study by Hardy et al [24], non-selective (ibuprofen) and selective (valeryl salicylate, DuP697 and NS398) COX-1 and COX-2 inhibitors were given intravenously to monitor changes in the concentrations of PGE2 and PGF2a and also changes in prostaglandin E (EP) and prostaglandin F (FP) receptors in retinal vasculature in newborn pigs. The results showed reductions in the concentrations of PGE2 and PGF2a in conjunction with increases in EP and FP receptors. Li et al also demonstrated increases in EP and FP receptor densities in brain synaptosomes in newborn pigs after treatment with ibuprofen or indomethacin [25]. Therefore, exogenous PGs in glaucoma patients may potentiate the IOP reduction effect through an increase in FP receptor expression following NSAID treatment.

Moreover, Maihöfner et al reported that patients with primary open-angle glaucoma (POAG) and steroid-induced glaucoma tended to lose COX-2 expression in the nonpigmented secretary epithelium of the ciliary body compared with normal eyes [26]. There is a general consensus that COX-2 plays an important role in PG formation. This may imply that POAG patients have reduced endogenous PG formation with subsequent FP receptor upregulation,

and this may be more obvious in glaucomatous eyes because FP receptors are over-expressed in glaucomatous tissue [27]. This explanation may be similar to that proposed by Costagliola with regards to the upregulation of receptors after the addition of topical NSAIDs to PG analogues [22].

The results of two studies in our meta-analysis contradict our hypothesis. Chiba [14] reported that an elevated IOP was noted 4–6 weeks after applying topical NSAIDs to topical PG analogues compared to a control group (topical PG analogues only). However, the increase in IOP between these two groups did not reach statistical significance. Moreover, a trend of decreasing IOP was found in both groups after the 6th week. In addition, R. Sorkhabi [16] et. al. reported a statistically significant increase in IOP was only found in the 2nd week after adding topical NSAIDs to topical PG analogues. Moreover, a decrease in IOP was noted beyond 2 weeks. These contradicting results may be due to racial differences, as Asian people have a higher rate of non-response to PG analogues than European or American people [28]. The reason may due to a higher uveoscleral flow in Caucasians when comparing with Asian populations, and thus a lower response rate to uveoscleral drugs such as prostaglandins may be found among Asians [29]. However, the underlying mechanism is still unclear, and further studies are needed to investigate this issue.

There are several limitations to this study, including the small number of enrolled studies. Only seven randomized trials were found and included in this meta-analysis. Although our analysis achieved enough statistical power, the publication bias may exist in our study [30]. In addition, different kinds of NSAIDs and PG analogues were used in different studies. Although there were some subtle differences in applying different PG analogues clinically, there were too few studies to categorize them into different groups for comparison. Moreover, our findings may only be applicable to glaucoma patients, as there were no differences in IOP between the normal subjects who received topical PG analogues alone and topical NSAIDs with PG analogues [20].

Moreover, systemic prostaglandins have been applied in cardiology for managing pulmonary hypertension [31], in obstetrics for inducing childbirth or abortion [32], in urology for treating erectile dysfunction [33], in pediatrics for preventing closure of ductus arteriosus in newborns [34], and some other fields. Our analysis raises a question that if the synergic effect also exists in such systemic conditions. Hence, related research in other fields could be done in the future.

In conclusion, this meta-analysis suggests that topical NSAIDs may enhance the IOP lowering effect of topical PG analogues in glaucoma patients. Therefore, the short-term usage of topical NSAIDs may not be contraindicated in glaucoma patients receiving PG analogues to control IOP.

## Supporting information

**S1 Checklist. PRISMA checklist.**
(DOC)

## Acknowledgments

We would like to thank the Biostatistics Task Force of Taipei Veterans General Hospital for helping us with the statistical analysis and STATA.

## Author Contributions

**Conceptualization:** Kang-Jung Lo.

**Data curation:** Kang-Jung Lo.

**Formal analysis:** Kang-Jung Lo.

**Investigation:** Kang-Jung Lo.

**Supervision:** De-Kuang Hwang.

**Writing – original draft:** Kang-Jung Lo.

**Writing – review & editing:** Yu-Chieh Ko, De-Kuang Hwang, Catherine Jui-Ling Liu.

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
