## [Decision Letter · Decision Letter 0]

15 Jun 2020

PONE-D-20-12499

The influence of topical non-steroidal anti-inflammatory drugs on the intraocular pressure lowering effect of topical prostaglandin analogues - a systemic review and meta-analysis

PLOS ONE

Dear Dr. Hwang,

Thank you for submitting your manuscript to PLOS ONE. After careful consideration, we feel that it has merit but does not fully meet PLOS ONE’s publication criteria as it currently stands. Therefore, we invite you to submit a revised version of the manuscript that addresses the points raised during the review process.

Reviewer concerns were significant, particularly as pertains to the sample size. This might move you to place this in a specialty journal where it would get more interested readers. However, if a revision is attempted, carefully and fully address all concerns as detailed in reviews. It will also be important to make clear to readers that there are not many manuscripts to meta-analyze thus limiting the interpretation. Other points by reviewers are important and need to be addressed fully.

We look forward to receiving your revised manuscript.

Kind regards,

Ted S Acott, PhD

Academic Editor

PLOS ONE

Additional Editor Comments:

Due to the small sample size, not many studies to meta-analyze, one reviewer was less enthusiastic. This might be better in a specialty journal where it would be read? Both reviewers had significant concerns that need to be carefully and completely addressed. If a revision is attempted, it will be important to make clear that this is a very small samples size, which limits interpretation.

Journal Requirements:

Reviewers' comments:

Reviewer's Responses to Questions

**Comments to the Author**

1. Is the manuscript technically sound, and do the data support the conclusions?

Reviewer #1: Yes

Reviewer #2: Partly

2. Has the statistical analysis been performed appropriately and rigorously? 

Reviewer #1: Yes

Reviewer #2: I Don't Know

3. Have the authors made all data underlying the findings in their manuscript fully available?

Reviewer #1: Yes

Reviewer #2: No

4. Is the manuscript presented in an intelligible fashion and written in standard English?

Reviewer #1: Yes

Reviewer #2: Yes

5. Review Comments to the Author

Reviewer #1: The authors present a meta-analysis of the IOP outcomes from use of PGAs and NSAIDs. The manuscript is technically sound but suffers from a low number of available articles to use on the analysis. Furthermore, more than half the studies had unclear methods of blinding in respective RCTs. This makes a good meta-analysis difficult. Furthermore, this manuscript would be more appropriate for a specialty journal.

1) Although no PGA/NSAID studies on Unoprostone, Vyzulta, or flurbiprofen may exist it would be more complete to search for those as well.

2) Results section : "...NSAIDS were evacuated." evacuated should be "excluded. "

3) Figures 2 and 3 have no legends. Each should have a specific legend that describes the data listed and what the SMD is. I assume that this is IOP.

Overall, a nice manuscript with an important concept the NSAIDs can be used with PGAs. However, there are just very few RCTs to subject to a meta-analysis. This could be of interest to a Glaucoma specialty journal.

Reviewer #2: It may be beneficial for the readers to include the actual IOP change rather than just the SMD. This ca be incorporated either in Table 2 or on the side of the Forest plot itself.

A comment on the magnitude of difference to be expected with the addition of NSAID to a prostaglandin with be clinically useful. What is the unit for the calculated SMD of 1.64; is it mmHg? This can be included in the table and the IOP outcome paragraph.

Simin study (discussion para 2, quoted reference 7) sems a bit problematic. For one, it does not match with reference 7 in the list of references and I am unable to locate it anywhere else in the references to look it up. Additionally, it seems that the prostaglandin group which by the description provided is just a recheck after not adding anything/adding placebo has a SMD of 2.55 in that group. Is there any explanation in the paper for this IOP lowering? When compared to the NSAID added group with a SMD of 2.93 and significant overlap in confidence interval, I’m not sure the study fits the preceding statement of “enhanced IOP lowering effect was reported when adding NSAIDs to PG analogues”. Please review the SMD numbers for accuracy or perhaps consider revising the interpretation.

Page 13, end of para 3: A higher uveoscleral flow in Caucasians and higher trabecular flow in Chinese (Asian) populations may be related to lower response rates of uveoscleral drugs such as prostaglandins in Asians (Fan et al, BJO Dec 2019). Authors may consider including the reference as a possible explanation for the difference.

6. PLOS authors have the option to publish the peer review history of their article (what does this mean?). If published, this will include your full peer review and any attached files.

Reviewer #1: No

Reviewer #2: No

---

## [Author Response · Author response to Decision Letter 0]

31 Jul 2020

Dear editor:

Thank you for considering our work entitled “The influence of topical non-steroidal anti-inflammatory drugs on the intraocular pressure lowering effect of topical prostaglandin analogues - a systemic review and meta-analysis”. We really appreciate the time and effort you and reviewers have dedicated to providing insightful feedback on ways to strengthen our paper. We have incorporated changes that reflect the detailed suggestions you and the reviewer have graciously provided. Following are the point-by-point responses to each reviewer’s comments.

Additional Editor Comments:

Due to the small sample size, not many studies to meta-analyze, one reviewer was less enthusiastic. This might be better in a specialty journal where it would be read? Both reviewers had significant concerns that need to be carefully and completely addressed. If a revision is attempted, it will be important to make clear that this is a very small samples size, which limits interpretation.

Response: 

Thanks for your valuable comment and reminding. Prostaglandin analogues and NSAID are two of the most commonly used eyedrops in clinical practice of ophthalmology. The former is a crucial anti-glaucomatous medication and the latter is for preventing/treating macular edema after ocular surgery. More than half of the ophthalmologists have experience in these two drugs. Since the underlying mechanism of these two drugs are theoretically opposite, there were some debates regarding whether we should avoid NSAID or discontinue PG if a glaucoma patient suffers from macular edema after surgery. Patient’s intraocular pressure could become relatively hard to be controlled if we discontinue PG. On the other hand, long-term macular edema would result in visual decline in patients as well. Although the clinical trials regarding this topic are scarce, we really think it is important and significant to clarify and highlight the synergic effect of these two medications so that ophthalmologist doesn’t have to avoid or discontinue them in glaucoma patients who need both of the medications.

 Besides, although this meta-analysis focused on topical NSAIDs and PG analogues in ophthalmology, we could also draw attention in the other fields. For example, systemic prostaglandins have been applied in cardiology for managing pulmonary hypertension, in obstetrics for inducing childbirth or abortion, in urology for treating erectile dysfunction, in pediatrics for preventing closure of ductus arteriosus in newborns, and some other fields. It is also debatable if we should avoid systemic NSAIDs in these patients. Our analysis raises a question that if the synergic effect also exists in such systemic conditions. More clinical or experimental studies in the other fields might be inspired by our results. Therefore, we sincerely hope our work could be published in a famous comprehensive journal such as PLoS ONE and share our results to readers in every fields. We’ve added some clinical significance in the manuscript. (Line 315 to 321, page 9-10) 

 We agreed that the sample size is small in this study. We’ve modified the DISCUSSION part and highlight this limitation in the manuscript in accordance to your suggestions. (Line 303 to 305, page 9)

Journal Requirements:

Response: 

 We already rewrote our manuscript to meet the requirement of this journal, and we apologized for our mistake on the sentence “data from this study are available upon request”. Since this is a meta-analysis, all data were available to the public. 

Reviewer 1’s comments:

The authors present a meta-analysis of the IOP outcomes from use of PGAs and NSAIDs. The manuscript is technically sound but suffers from a low number of available articles to use on the analysis. Furthermore, more than half the studies had unclear methods of blinding in respective RCTs. This makes a good meta-analysis difficult. Furthermore, this manuscript would be more appropriate for a specialty journal.

Response: Thank for your valuable comments. As we’ve answered the editor’s comment previously in this letter, “Prostaglandin analogues and NSAID are two of the most commonly used eyedrops in clinical practice of ophthalmology. The former is a crucial anti-glaucomatous medication and the latter is for preventing/treating macular edema after ocular surgery. More than half of the ophthalmologists have experience in these two drugs. Since the underlying mechanism of these two drugs are theoretically opposite, there were some debates regarding whether we should avoid NSAID or discontinue PG if a glaucoma patient suffers from macular edema after surgery. Patient’s intraocular pressure could become relatively hard to be controlled if we discontinue PG. On the other hand, long-term macular edema would result in visual decline in patients as well. Although the clinical trials regarding this topic are scarce, we really think it is important and significant to clarify and highlight the synergic effect of these two medications so that ophthalmologist doesn’t have to avoid or discontinue them in glaucoma patients who need both of the medications.

 Besides, although this meta-analysis focused on topical NSAIDs and PG analogues in ophthalmology, we could also draw attention in the other fields. For example, systemic prostaglandins have been applied in cardiology for managing pulmonary hypertension, in obstetrics for inducing childbirth or abortion, in urology for treating erectile dysfunction, in pediatrics for preventing closure of ductus arteriosus in newborns, and some other fields. It is also debatable if we should avoid systemic NSAIDs in these patients. Our analysis raises a question that if the synergic effect also exists in such systemic conditions. More clinical or experimental studies in the other fields might be inspired by our results.”

 Although the sample size is relatively small in this meta-analysis, they still provide enough statistical power (Journal of Educational and Behavioral Statistics 2010, 35(2), 215–247.) On the other hand, we agreed with you that the low number and detail information of available RCTs is a main drawback in this study that we should carefully interpret the results. We’ve modified the DISCUSSION part and highlight this limitation in the manuscript in accordance to your comment. (Line 303 to 305, page 9)

1. Although no PGA/NSAID studies on Unoprostone, Vyzulta, or flurbiprofen may exist it would be more complete to search for those as well.

Response: Thank you very much for this suggestion. We’ve re-performed the literature review adding unoprostone, Vyzulta, and flurbiprofen as keywords. Unfortunately, no additional clinical trial was found based on the new criteria. We’ve revised the Methods in the manuscript with this change in searching criteria. (Line 113-117, page 3) 

2. Results section : "...NSAIDS were evacuated." evacuated should be "excluded. "

Response: Thanks for your reminding. We’ve corrected the typo. (Line 173, Page 5) 

3. Figures 2 and 3 have no legends. Each should have a specific legend that describes the data listed and what the SMD is. I assume that this is IOP.

Response: We’ve added legends into figure 2 and 3 in accordance to your comment. (Line 217-230, page 7). SMDs in the figure represent the standardized mean difference of IOP (mmHg), which were calculated with Cohen method. We apologized for making the confusion. We’ve added the footnotes in the figure and the explanation in the manuscript. (Line 215-230, page 7; figure 2; and figure 3) 

Overall, a nice manuscript with an important concept the NSAIDs can be used with PGAs. However, there are just very few RCTs to subject to meta-analysis. This could be of interest to a Glaucoma specialty journal. 

Response: We really appreciate your valuable comments and considerations. As we’ve answered in the previous comments, we think the statistical power of this meta-analysis was enough despite the small number of available trials. We sincerely hope our work could be published in PLoS ONE since the results of this study might inspire many ophthalmologists and medical physicians in the other field as well. 

Reviewer 2’s comments:

It may be beneficial for the readers to include the actual IOP change rather than just the SMD. This can be incorporated either in Table 2 or on the side of the Forest plot itself.

Response: Thanks. We’ve included the information of IOP in baseline and after treatments in each trial and incorporated these data into figure 2 and figure 3 in accordance to your suggestions. 

A comment on the magnitude of difference to be expected with the addition of NSAID to a prostaglandin with be clinically useful. What is the unit for the calculated SMD of 1.64; is it mmHg? This can be included in the table and the IOP outcome paragraph.

Response: The unit of the SMD (standardized mean difference) in the results are mmHg. Our results showed that adding topical NSAID on prostaglandin analogue may additionally decrease 1.3mmHg of IOP in patients with glaucoma. We apologized for the confusion and lack of the explanation in the original manuscript. We’ve added the explanation in the table and manuscript in accordance to your suggestion. “The unit of SMD is in mmHg.“ (Line 223 and 230, Page 7)

Simin study (discussion para 2, quoted reference 7) sems a bit problematic. For one, it does not match with reference 7 in the list of references and I am unable to locate it anywhere else in the references to look it up. Additionally, it seems that the prostaglandin group which by the description provided is just a recheck after not adding anything/adding placebo has a SMD of 2.55 in that group. Is there any explanation in the paper for this IOP lowering? When compared to the NSAID added group with a SMD of 2.93 and significant overlap in confidence interval, I’m not sure the study fits the preceding statement of “enhanced IOP lowering effect was reported when adding NSAIDs to PG analogues”. Please review the SMD numbers for accuracy or perhaps consider revising the interpretation.

Response: Thank you so much for this valuable comment. We carefully reviewed the methods of Simin’s study again in accordance to your comment and found that the original “baseline” IOPs in this study were measured before PGE treatments, which was markedly different from the other studies, in which the “baseline” IOPs were measured under PGE therapy right before adding the NSAID. Hence the original results of Simin’s study should not directly compare with the others. 

To overcome this fundamental problem, we’ve modified patients’ baseline IOPs in Simin’s study to the IOPs measured after 4 weeks of prostaglandin monotherapy and re-done the meta-analysis again. Fortunately, the significantly synergic effect was still observed in the combine therapy group, with only a little bit decrease of overall standardized mean difference (from 1.64 to 1.33 mmHg). The overall standardized mean difference in PGE monotherapy remained non-significance (from 0.61 to -0.03 mmHg). 

 We apologize for the mistake and have already revised the manuscript and figure based on these new results.

Page 13, end of para 3: A higher uveoscleral flow in Caucasians and higher trabecular flow in Chinese (Asian) populations may be related to lower response rates of uveoscleral drugs such as prostaglandins in Asians (Fan et al, BJO Dec 2019). Authors may consider including the reference as a possible explanation for the difference.

Response: Thank you for the suggestion. We’ve added this valuable reference in our manuscript in accordance to your comment. (Line 295-298, page 9) 

Again, thank you very much for considering our manuscript through Reviewer’s most valuable comments. We have worked hard to answer each query and accordingly have carefully made revisions. We hope that this revised version is qualified to be accepted for publication.

Sincerely,

De-Kuang Hwang

---

## [Decision Letter · Decision Letter 1]

2 Sep 2020

The influence of topical non-steroidal anti-inflammatory drugs on the intraocular pressure lowering effect of topical prostaglandin analogues - a systemic review and meta-analysis

PONE-D-20-12499R1

Dear Dr. Hwang,

We’re pleased to inform you that your manuscript has been judged scientifically suitable for publication and will be formally accepted for publication once it meets all outstanding technical requirements.

Kind regards,

Ted S Acott, PhD

Academic Editor

PLOS ONE

Additional Editor Comments (optional):

Reviewers' comments:

Reviewer's Responses to Questions

**Comments to the Author**

1. If the authors have adequately addressed your comments raised in a previous round of review and you feel that this manuscript is now acceptable for publication, you may indicate that here to bypass the “Comments to the Author” section, enter your conflict of interest statement in the “Confidential to Editor” section, and submit your "Accept" recommendation.

Reviewer #1: All comments have been addressed

2. Is the manuscript technically sound, and do the data support the conclusions?

Reviewer #1: Yes

3. Has the statistical analysis been performed appropriately and rigorously? 

Reviewer #1: Yes

4. Have the authors made all data underlying the findings in their manuscript fully available?

Reviewer #1: Yes

5. Is the manuscript presented in an intelligible fashion and written in standard English?

Reviewer #1: Yes

6. Review Comments to the Author

Reviewer #1: This is a revised manuscript presenting a meta-analysis of the IOP effects of PGAs with concurrent NSAID use. Overall, an improved manuscript in which the authors have addressed prior comments. Concurrent of these 2 classes of medications doesn't seem to adversely affect IOP.

7. PLOS authors have the option to publish the peer review history of their article (what does this mean?). If published, this will include your full peer review and any attached files.

Reviewer #1: No

---

## [Editor Report · Acceptance letter]

4 Sep 2020

PONE-D-20-12499R1 

The influence of topical non-steroidal anti-inflammatory drugs on the intraocular pressure lowering effect of topical prostaglandin analogues - a systemic review and meta-analysis 

Dear Dr. Hwang:

I'm pleased to inform you that your manuscript has been deemed suitable for publication in PLOS ONE. Congratulations! Your manuscript is now with our production department. 

Kind regards, 

on behalf of

Dr. Ted S Acott 

Academic Editor

PLOS ONE